# Ligand recognition and biased agonism of the D1 dopamine receptor

Xiao Teng [1,2], Sijia Chen[2,3], Yingying Nie[2], Peng Xiao [4], Xiao Yu [5], Zhenhua Shao [6] & Sanduo Zheng [1,2,3 ✉]

Dopamine receptors are widely distributed in the central nervous system and are important therapeutic targets for treatment of various psychiatric and neurological diseases. Here, we report three cryo-electron microscopy structures of the D1 dopamine receptor (D1R)-Gs complex bound to two agonists, fenoldopam and tavapadon, and a positive allosteric modulator LY3154207. The structure reveals unusual binding of two fenoldopam molecules, one to the orthosteric binding pocket (OBP) and the other to the extended binding pocket (EBP). In contrast, one elongated tavapadon molecule binds to D1R, extending from OBP to EBP. Moreover, LY3154207 stabilizes the second intracellular loop of D1R in an alpha helical conformation to efficiently engage the G protein. Through a combination of biochemical, biophysical and cellular assays, we further show that the broad conformation stabilized by two fenoldopam molecules and interaction between TM5 and the agonist are important for biased signaling of D1R.

[1] Tsinghua Institute of Multidisciplinary Biomedical Research, Tsinghua University, Beijing, China. [2] National Institute of Biological Sciences, Beijing, China. [3] Graduate School of Peking Union Medical College, Beijing, China. [4] Key Laboratory Experimental Teratology of the Ministry of Education and Department of Biochemistry and Molecular Biology, School of Basic Medical Sciences, Cheeloo College of Medicine, Shandong University, Jinan, Shandong, China. [5] Key Laboratory Experimental Teratology of the Ministry of Education and Department of Physiology, School of Basic Medical Sciences, Cheeloo College of Medicine, Shandong University, Jinan, Shandong, China. [6] Division of Nephrology and Kidney Research Institute, State Key Laboratory of Biotherapy and Cancer Center, West China Hospital, Sichuan University, Chengdu, Sichuan, China. ✉email: zhengsanduo@nibs.ac.cn

Dopamine was discovered as a neurotransmitter in the central nervous system (CNS) in 1957 by refs. [1,2]. In subsequent decades, functional interrogation of dopamine revealed its important roles in regulating multiple physiological processes such as motivation, cognition, locomotor activity, hypertension, and endocrine regulation[3–6]. Dopamine acts on five distinct G protein-coupled dopamine receptors (DRs) expressed on the plasma membrane to initiate downstream cellular signaling and fulfill its biological function. Among the DRs, D1-like receptors (D1R and D5R) primarily signal through the stimulatory G protein subunit ($G\alpha_s$) and stimulate cyclic AMP (cAMP) production, whereas D2-like receptors (D2R, D3R, and D4R) engage the inhibitory G protein subunit ($G\alpha_{i/o}$) and inhibit cAMP production[3,4]. D1-like receptors are exclusively expressed in postsynaptic neurons, whereas D2-like receptors are expressed in both presynaptic and postsynaptic neurons.

The dopaminergic system has been one of the most extensively studied neurotransmitter systems over the past few decades, mainly because dopaminergic dysfunction has been implicated in multiple psychiatric and neurological diseases including Parkinson's disease (PD), attention deficit hyperactivity disorder, and schizophrenia[7–9]. Basic research on dopamine has greatly enhanced our understanding of the molecular and cellular bases of these diseases and contributed to the development of modern therapeutics against them[3]. Most antipsychotic drugs targeting DRs for treating schizophrenia and drugs for treating PD are based on D2-like receptor antagonism and agonism, respectively[3]. In contrast, fenoldopam, which is used for the treatment of severe hypertension, is the only marketed D1R-like selective agonist[10]. Since it has been suggested that D1R is highly expressed in the striatum and prefrontal cortex, and is involved in movement and cognition, pharmacologists and the pharmaceutical industry have been striving to develop D1R-subtype selective agonists to alleviate movement disorders and cognitive impairment associated with PD[11–14]. However, catechol-based agonists of D1R failed to meet clinical goals because of their poor drug-like properties, inverted U-shaped dose-response curves, and induction of tachyphylaxis due to receptor desensitization[11,15,16]. To circumvent these obstacles, non-catechol agonists and positive allosteric modulators (PAMs) have been developed[11,12,17–20], some of which including tavapadon and LY3154207 have shown promising safety and efficacy in clinical trial studies[21,22].

Structural approaches including nuclear magnetic resonance, X-ray crystallography, and cryo-electron microscopy (cryo-EM) in combination with molecular dynamics simulations have been widely used to understand the molecular mechanisms underlying ligand recognition and selectivity and receptor activation by agonists. Because most dopaminergic drugs target D2-like receptors, they have been the subject of intensive structural studies in recent years. To date, crystal structures of antagonist-bound D2R, D3R, and D4R and cryo-EM structures of agonist-bound D2R and D3R in complex with the $G_i$ heterotrimer have been reported[23–30]. Until recently, several groups have solved structures of D1R-$G_s$ bound to catechol agonists, non-catechol agonists, and a PAM[29,31–33]. Interestingly, the binding pose and receptor interactions of ligands including dopamine and a catechol agonist SKF83959 were slightly different between two of the studies[34]. The structures of D1R-$G_s$ bound to the non-catechol agonists PW0464 and Compound 1 revealed that the interactions and mechanism of receptor activation of non-catechol agonists are distinct from those of catechol agonists. While PW0464 shows biased signaling for G protein pathway, a structurally similar drug tavapadon can activate the G protein pathway as well as the arrestin pathway which is associated with receptor desensitization[17,18].

In this study, we determine cryo-EM structures of the human D1R-$G_s$ complex bound to fenoldopam, tavapadon, and LY3154207.

Two fenoldopam molecules bind in the orthosteric binding pocket (OBP) and extended binding pocket (EBP), respectively. The one in the EBP stabilizes the receptor in a broad conformation that is necessary for efficient β-arrestin coupling. Increasing the distance between TM5 and tavapadon through mutagenesis enhances the efficiency of tavapadon for arrestin coupling while reducing its efficiency for G protein coupling. LY3154207 enhances binding affinity of the agonist by stabilizing the second intracellular loop (ICL2) in an alpha-helical structure and increasing the proportion of the receptors adopting active conformation. Taken together, this work provides insights into the molecular basis of ligand recognition, biased agonism and allosteric modulation for D1R.

## Results

**Overall structure of the D1R-$G_s$ complex.** In a companion manuscript[35], we present cryo-EM structures of the dopamine-bound D1R-$G_s$ protein complex in the nucleotide-free and the nucleotide-bound state using a GPCR-truncated $G\alpha_s$ (mini-$G_s$) fusion protein strategy. These structures reveal intermediate states of GPCR-G protein coupling. Here, using the same expression and purification strategy, we determined three cryo-EM structures of the activated D1R-min-$G_s$-Nb35 complex bound to a catechol agonist, fenoldopam; the complex bound to a non-catechol agonist, tavapadon; and the complex simultaneously bound to a PAM LY3154207, and endogenous dopamine, at a global resolution of 3.2, 3.3, and 3.0 Å, respectively (Fig. 1 and Supplementary Table 1). Owing to the high quality of cryo-EM maps, all the side chains of most residues in D1R, the $G_s$ heterotrimer, and Nb35, except the extreme terminus and some internal loops of D1R, were well defined. Moreover, the aforementioned ligands could be modeled into the density map with high confidence. D1R-mini-$G_s$-Nb35 adopts a conformation similar to that of the recently published crystal structure of D1R bound to the full-length $G\alpha_s$ with an overall root-mean-square deviation (RMSD) of 0.8 Å over 641 Cα atoms (Supplementary Fig. 1). The interaction interface of the receptor and G protein are nearly identical, whereas the conformation of the receptors near the ligand-binding site differs significantly from the crystal structure; this is likely due to the N-terminal T4L fusion protein to the receptor, which was used to facilitate crystallization. Our cryo-EM structures share some common features including a longer helical extension of TM5 in D1R and a different orientation of the $G_s$ heterotrimer relative to the receptor compared with the β2 adrenergic receptor (β2AR)-$G_s$ complex[36]. These results suggest that the mini-$G_s$ and the fusion strategy had little effect on the structural assembly[29,31–33]. Overall, the structures of fenoldopam-, tavapadon-, and dopamine-LY3154207-bound D1R-mini-$G_s$-Nb35 are similar to the structure of D1R-mini-$G_s$-Nb35 bound only to dopamine, with RMSD values of 0.6, 0.7, and 0.9 Å, respectively, over the Cα atoms of the entire complex.

**Two fenoldopam molecules in the receptor binding pocket.** Unassigned EM density spanning from OBP to EBP was observed after modeling all protein components in the cryo-EM map of the fenoldopam-bound complex. Surprisingly, two fenoldopam molecules could be unambiguously modeled into the density (Fig. 2a and Supplementary Fig. 2). One molecule is located in the OBP and the other in the EBP, and they interact with each other through π–π interaction between the phenol groups (Fig. 2b). The fenoldopam in the OBP engages in similar interactions as dopamine, including salt bridge interactions between D103$^{3.32}$ (Ballesteros–Weinstein numbering) and the amine group, hydrogen bond interactions between the double hydroxyl groups of catechol and S198$^{5.42}$ and N292$^{6.55}$, as well as hydrophobic contacts involving the catechol ring and surrounding hydrophobic residues such as F288$^{6.51}$ and

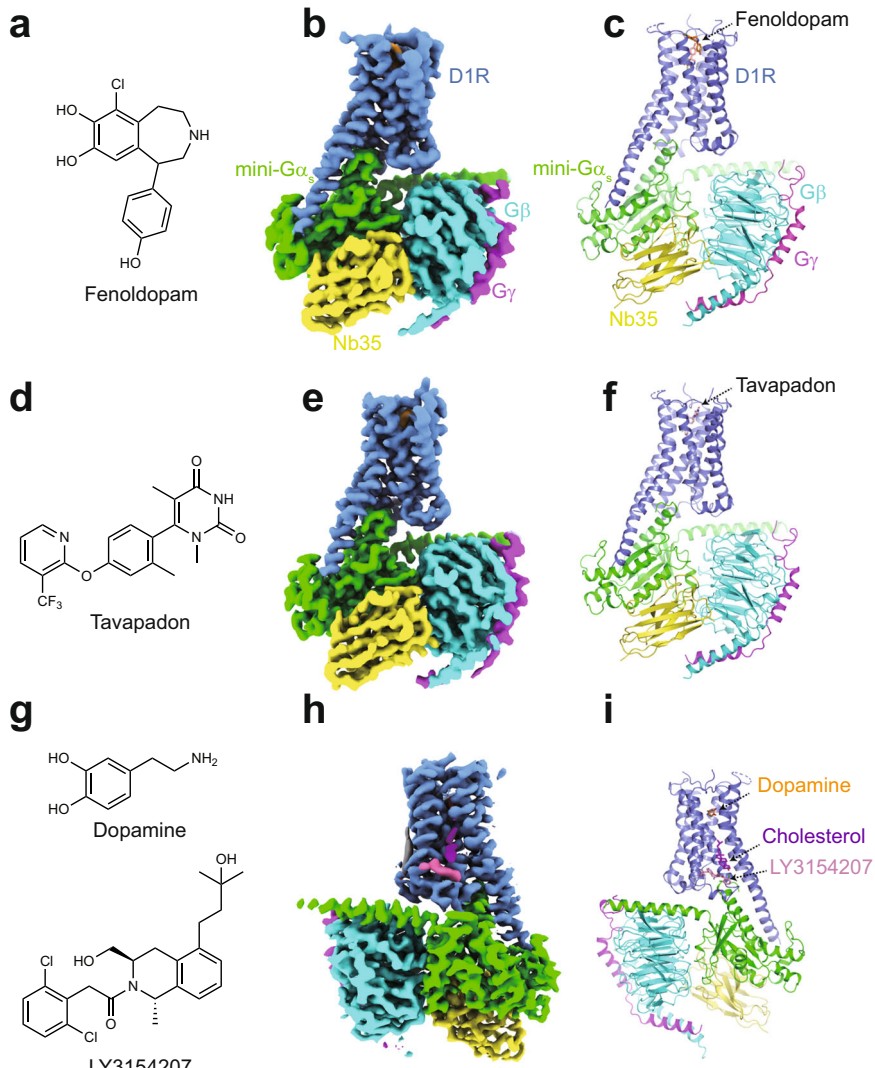

**Fig. 1 Overall structures of the D1R-mini-Gs-Nb35 complexes. a–c** Structure of the fenoldopam-bound D1R complex. **a** Chemical structure of fenoldopam. **b** Cryo-EM map of the complex. **c** Ribbon representations of the structure. D1R, mini-Gα_s, Gβ, Gγ, and Nb35 are colored in blue, green, cyan, magenta, and yellow, respectively. **d–f** Structure of the tavapadon-bound D1R complex. **g–i** Structure of the dopamine-LY3154207-bound D1R complex.

F289[6.52] (Fig. 2b). The phenol group projects into the EBP underneath ECL2 and is surrounded by hydrophobic residues including L190[ECL2] and F313[7.35] and the phenol group of the neighboring fenoldopam. The other fenoldopam resides in the EBP pocket, with the benzazepine group located at the extracellular vestibule and the phenol group facing toward the OBP. The amino group of the benzazepine forms strong polar interactions with K81[2.61] and D314[7.36] (Fig. 2b). Moreover, the two hydroxyl groups of the catechol moiety engage in hydrogen bond interactions with the main chain carbonyl group of S188[ECL2] and the side chain hydroxyl group of S189[ECL2]. This fenoldopam molecule is further stabilized by hydrophobic interactions among the phenol group, W99[3.28], V317[7.39], and the aliphatic part of K81[2.61] and a hydrogen bond interaction between the hydroxyl group of the phenol and D103[3.32].

To support our structural findings, we performed both NanoBiT mini-G_s recruitment and cAMP accumulation assays to analyze the effects of mutations in the ligand-binding pocket on D1R activation by the agonist. Consistent results were obtained using the two assays (Fig. 2c, d), further supporting the fact that mini-G_s can recapitulate the full-length G_s in terms of studying GPCR activation and coupling selectivity[37]. As

expected, mutation of residues involved in binding fenoldopam in the OBP such as S198[5.42], N292[6.55], F288[6.51], F289[6.52], W321[7.43], and L190[ECL2] significantly impaired the potency of fenoldopam, and the D103V mutation led to a complete loss in fenoldopam potency (Fig. 2c, d). While the S202[5.46] mutation significantly reduced the dopamine potency[32,33,35], it had little effect on the fenoldopam potency because of the longer distance between the hydroxyl group and S202[5.46].

In contrast to our structure, a recently published fenoldopam-bound D1R-G_s structure revealed only one fenoldopam in the OBP[33]. When superimposing the two structures, the entire G_s heterotrimer moves toward the receptor by ~2.5 Å compared with ours, and the cytoplasmic ends of TM5, TM6, and TM7 move inwards, resulting in a narrower intracellular cleft of the receptor (Fig. 2e). Specifically, the upward movement of the G protein leads to the upward movement of the side chains of residues in TM3 and TM5 as indicated by the movement of DRY motif and S202[5.46], respectively, which causes the subsequent upward movement of fenoldopam in the OBP (Supplementary Fig. 3a, b). Moreover, the inward movement of TM6 allosterically induces conformational changes in the toggle switch W285[6.48], the residue F281[6.44] in the PIF motifs, and residues near the fenoldopam in the

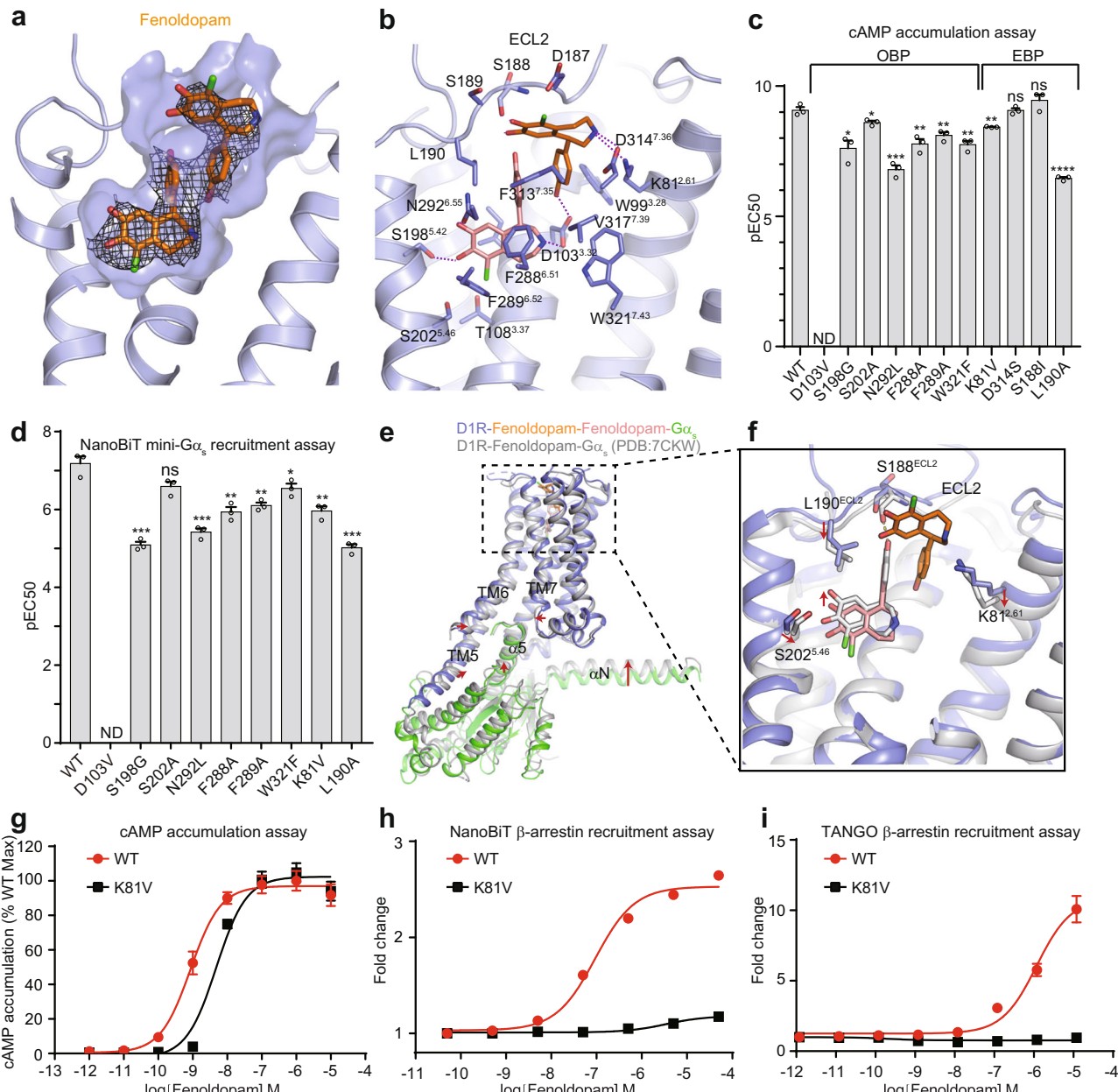

**Fig. 2 Distinct binding mode of fenoldopam in D1R. a** Surface view of the fenoldopam-binding pocket. EM density map for fenoldopam is shown. **b** Detailed interactions between the two fenoldopam molecules and the receptor. Dashed lines represent hydrogen bonds. **c** cAMP accumulation assay for D1R wild-type (WT) and mutants activated by fenoldopam. The assay was performed in HEK293 cells stably expressing the cAMP GloSensor. EC50 values are determined from three independent experiments. Data are presented as mean values ± SEM. Statistical significances of D1R mutants were obtained using two-tailed Student's $t$ test (*$p < 0.05$, **$p < 0.01$, ***$p < 0.001$, ****$p < 0.0001$). ND not determined, ns no significant difference. The exact $p$ values are as follows, S198G, *$p = 0.0104$; S202A, *$p = 0.0265$; N292L, ***$p = 0.0004$; F288A, **$p = 0.0042$; F289A, **$p = 0.0048$; W321F, **$p = 0.002$; K81V, **$p = 0.0059$; D314S, $p = 0.9475$; S188I, $p = 0.2013$; L190A, ****$p = 0.00003$. **d** NanoBiT mini-Gαs recruitment assay for D1R WT and mutants. Data are presented as mean values ± SEM from three independent experiments. $p$ values were obtained by two-tailed Student's $t$ test (*$p < 0.05$, **$p < 0.01$, ***$p < 0.001$, ****$p < 0.0001$). The exact $p$ values are as follows, S198G, ***$p = 0.0005$; S202A, $p = 0.0539$; N292L, ***$p = 0.001$; F288A, **$p = 0.0048$; F289A, **$p = 0.0052$; W321F, *$p = 0.042$; K81V, **$p = 0.0047$; L190A, ***$p = 0.0004$. **e** Structural comparison of the fenoldopam-bound D1R complex between this study and previous studies (PDB ID: 7CKW) with the receptor aligned. **f** Close-up views of the conformational differences of the fenoldopam-binding pocket between different studies. Concentration-response curve for D1R WT and the K81V mutant activated by fenoldopam measured using cAMP accumulation assay (**g**), NanoBiT (**h**) and TANGO (**i**) β-arrestin recruitment assay. All data represent mean ± SEM from three independent experiments. Response of the cAMP assay is expressed as percentage of maximal response of WT. Source data are provided as a Source Date file.

EBP such as F313$^{7.53}$ (Supplementary Fig. 3b). The upward movement of fenoldopam in the OBP pulls down the ECL2 through hydrogen bond interaction between the hydroxyl group of phenol and the main chain carbonyl group of S188 on ECL2 (Fig. 2f). The resulting closer distance between ECL2 and fenoldopam in the OBP

generates a narrower space, which cannot accommodate the second fenoldopam in the EBP, accounting for its absence in the published structure. These conformational differences of the ligand-binding pocket and intracellular pocket of the receptor may arise from the different versions of Gαs in the complexes (mini-Gs versus the full-

length G protein), the different expression systems, and/or differences in purification strategy. Nevertheless, GPCRs should be considered as highly dynamic systems that exist in multiple active conformations[38,39]. Previous studies have shown that ligands with divergent signaling bias (Gq biased versus β-arrestin biased signaling) for angiotensin II type 1 receptor leads to the conformational heterogeneity of the receptor[40]. Therefore, we speculate that the additional fenoldopam in the EBP can stabilize a distinct conformation of the receptor, thereby leading to a different active state from that of D1R bound by only one fenoldopam in the OBP. Consistent with structural observations, microscale thermophoresis (MST) assay[41] shows that D1R accommodates two separate binding sites for fenoldopam (Supplementary Fig. 3c), while mutation of K81$^{2.61}$ that is involved in binding fenoldopam in the EBP (Fig. 2f) into the equivalent residue of D2R (valine) to minimize the effect of the mutation on receptor folding leads to one binding site for fenoldopam (Supplementary Fig. 3d). Moreover, K81$^{2.61}$V mutation almost completely abolished fenoldopam-induced β-arrestin recruitment as demonstrated by both NanoBiT[42] and TANGO[43] β-arrestin recruitment assays (Fig. 2h, i), but only reduced the potency of fenoldopam in G$_s$ activation by about 5-fold, which led to G protein biased signaling (Fig. 2g). These results indicate that the fenoldopam in the EBP likely stabilizes the receptor in a broad conformation that is important for efficient β-arrestin coupling. Interestingly, K81$^{2.61}$V mutation reduces the binding affinity of dopamine by about 15-fold (Supplementary Fig. 3e, f); it also significantly impairs the potency of dopamine in both G$_s$ activation and β-arrestin recruitment (Supplementary Fig. 3g, h). As K81$^{2.61}$ is distant from dopamine in OBP, it is not clear how K81$^{2.61}$V mutation affects the binding affinity as well as the potency of dopamine. One explanation could be that a second dopamine may bind in a similar way to fenoldopam in EBP, which is not captured in our structure likely due to its dynamic property. Further studies are required to clarify whether a second dopamine binding site exists in D1R. Taken together, these data suggest that the ligands are very dynamic in the binding pocket of receptors, and different ligand-binding modes could lead to the conformational heterogeneity of the intracellular pocket of the receptor, which may contribute to multiple active-state conformations and recruitment of various transducers.

**Receptor activation by non-catechol agonists**. In contrast to catechol agonists, non-catechol agonists have good CNS penetration, high selectivity, and superior pharmacokinetic profiles. Our structure reveals that a non-catechol agonist tavapadon occupies the pocket spanning from the OBP to the EBP (Fig. 3a and Supplementary Fig. 4). The trifluoromethyl-pyridine group of tavapadon at one end is located in the OBP, and engages in weaker polar interactions with S198$^{5.42}$ and D103$^{3.32}$ via its fluorine and nitrogen atoms, respectively (Fig. 3b). As a result, less significant inward movement of S198$^{5.42}$, S202$^{5.46}$, and P206$^{5.50}$ from the PIF motif is observed (Fig. 3c), and the D103$^{3.32}$ mutation reduced the potency of tavapadon by about ten-fold (Fig. 3d), whereas the same mutation led to a complete loss of potency for fenoldopam (Fig. 2c). Moreover, the S198$^{5.42}$G mutation influences the G$_s$ coupling efficiency of tavapadon to a much lesser extent than that of catechol agonists (Figs. 2c and 3d). Similar to the catechol ring of catechol agonists, the pyridine ring of tavapadon forms hydrophobic interactions with F288$^{6.51}$ and F289$^{6.52}$ in TM6 of D1R. Mutations in these residues significantly reduce the potency of tavapadon (Fig. 3d). A compound without the trifluoromethyl group in tavapadon can still activate the D1R, although its potency is weaker[12,44], suggesting that hydrophobic interactions between the compound and TM6 are essential for D1R activation. The connecting oxygen atom

potentially forms a hydrogen bond with the amide group of N292$^{6.55}$. Substitution of this oxygen with nitrogen would disfavor the hydrogen bond and reduced the potency in cAMP accoumulation[45]. To compensate the weak polar interactions in the OBP, tavapadon possesses a pyrimidine group at the other end, which extends to ECL2 and forms strong hydrogen bond interactions with main chains of C186$^{ECL2}$and S188$^{ECL2}$ and the side chain of K81$^{2.61}$ (Fig. 3b). L190$^{ECL2}$ plays an important role in stabilizing both catechol and non-catechol agonists through hydrophobic interactions. The central phenyl ring with a methyl group attached is surrounded by hydrophobic residues F288$^{6.51}$, F313$^{7.35}$, and L190$^{ECL2}$. Addition of a trifluoromethyl or fluorine group neighboring the methyl group in the central phenyl ring would cause a clash with the surrounding hydrophobic residues, accounting for their reduced potency in cAMP accumulation[45]. To further support our findings from structural analysis, we tested the effect of mutations of residues involved in binding the pyrimidine group of tavapadon on ligand potency (Fig. 3d). Mutations of K81$^{2.61}$ and L190$^{ECL2}$ significantly impaired the ligand potency. Consistent with previous studies[18], the S188A mutation did not affect ligand-binding, while the S188$^{ECL2}$L mutation reduced the potency of the ligand. As S188$^{ECL2}$ faces toward the solvent because of its hydrophilic nature, mutation of S188$^{ECL2}$ into a hydrophobic residue changes its main chain conformation and disrupts the hydrogen bond interaction between the main chain of S188$^{ECL2}$ and tavapadon. Superposition of the structures of fenoldopam- and tavapadon-bound D1R revealed that ECL2 adopts different conformations, indicating that ECL2 is quite flexible and can change its conformation in order to accommodate ligands with diverse scaffolds (Fig. 3c).

**Biased agonism**. Interestingly, tavapadon induces signaling through both the G protein and arrestin pathways, whereas PW0464, which has an additional oxygen atom between the fluoromethyl group and pyridine in tavapadon, shows a G protein bias[17]. As expected, comparison with a recently published structure of the PW0464-bound D1R revealed that the fluoromethyl group in the PW0464-bound D1R is closer to TM5 than that in the tavapadon-bound D1R (Fig. 3e, f)[33]. As a result, significant conformational changes of S198$^{5.42}$, S199$^{5.43}$, and S202$^{5.46}$ in TM5 are observed in the PW0464-bound D1R structure (Fig. 3f). The movement of ECL2 and the pyrimidine group between the tavapadon- and PW0464-bound structures is likely due to the conformational differences of the receptor and G protein interface (Fig. 3e, f), which also occur in the fenoldopam-bound structures (Fig. 2e). A similar effect has also been observed for the catechol agonist SKF83959, where an extra methyl group attached to the amine of the benzazepine in SKF81297 moves the double hydroxyl groups closer to TM5 and renders it unable to activate the β-arrestin pathway[46]. The common feature of GPCR activation is the contraction of the ligand-binding pocket, which is accompanied by outward movements of the cytoplasmic ends of TM5 and TM6[36,47]. Therefore, a closer distance between TM5 and the ligand may limit the contraction of the orthosteric pocket and reduce the degree of the overall signaling response. Indeed, both PW0464 and SKF83959 show slightly reduced potency in terms of G protein coupling but an almost complete loss of potency in the arrestin pathway, compared with tavapadon and SKF81297, respectively[17,48]. Similar to the steric effect due to the extra atom, addition of a chloride atom or methoxyl group near the trifluoromethyl group in tavapadon also leads to G protein biased signaling[12]. Therefore, β-arrestin coupling appears to be more sensitive to the close distance between ligands and TM5 than the G protein pathway. Moreover, in an extreme case, the antagonist SCH23390[49], in which one of the hydroxyl groups of

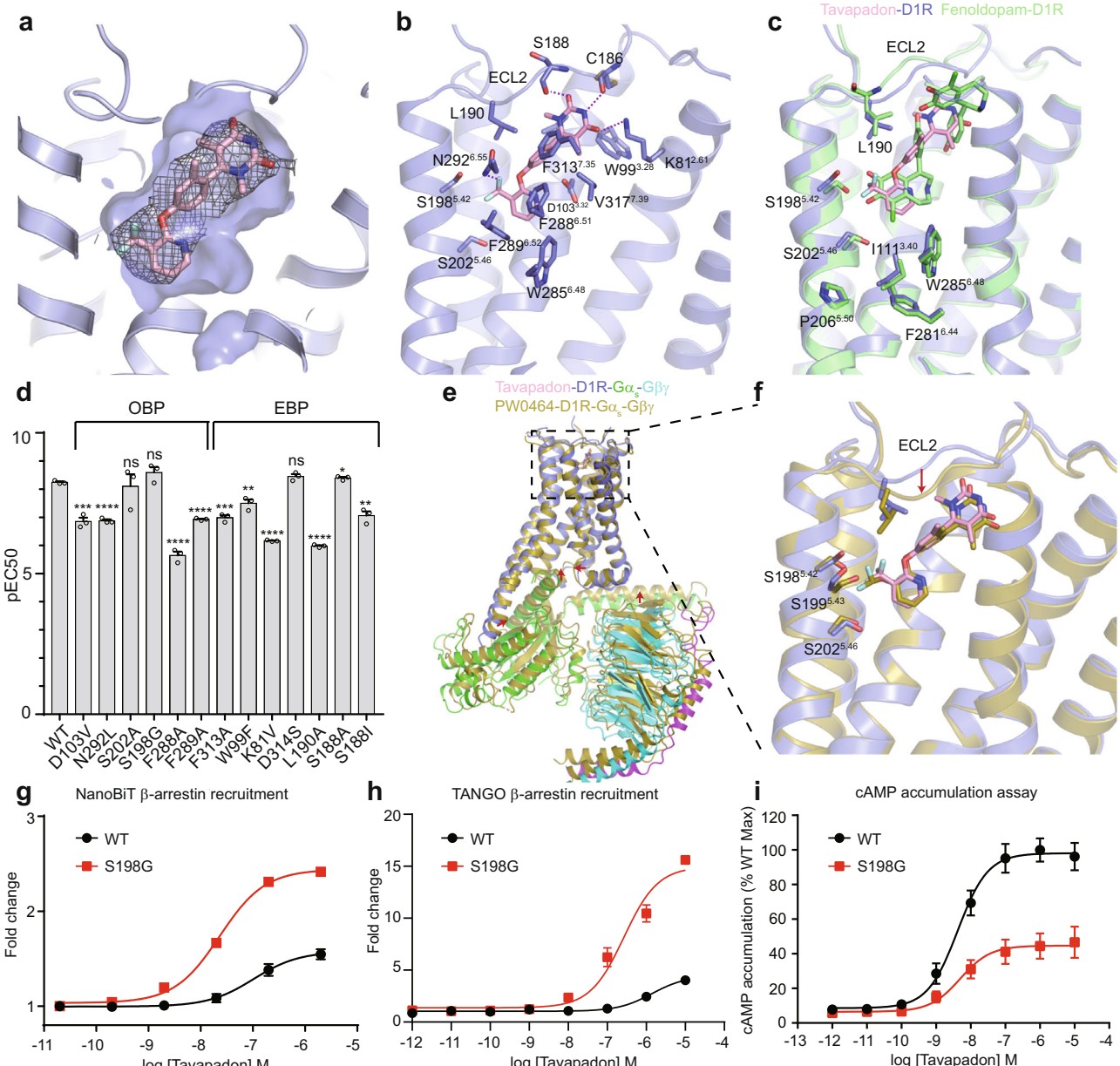

**Fig. 3 Non-catechol agonist recognition by D1R. a** Surface view of the tavapadon binding pocket. Tavapadon is shown as sticks, and EM density is shown for tavapadon. **b** Interactions between tavapadon and the receptor. **c** Structural comparison between the tavapadon- and fenoldopam-bound D1R. **d** Effect of D1R mutants on the ability of tavapadon to stimulate the production of cAMP. Three independent experiments were repeated for each construct. Data are presented as mean values ± SEM. $p$ values were calculated by means of two-tailed Student's $t$ test (*$p < 0.05$, **$p < 0.01$, ***$p < 0.001$, ****$p < 0.0001$). The exact $p$ values are as follows, D103V, ***$p = 0.0005$; N292L, ****$p = 0.00002$; S202A, $p = 0.7662$; S198G, $p = 0.1421$; F288A, ****$p = 0.000046$; F289A, ****$p = 0.000006$; F313A, ***$p = 0.0002$; W99F, **$p = 0.0055$; K81V, ****$p = 6.2E-07$; D314S, $p = 0.0785$; L190A, ****$p = 9.2E-07$; S188A, *$p = 0.0477$; S188I, **$p = 0.0015$. **e** Structural overlay of the tavapadon-bound D1R complex and PW0464-bound D1R complex; the receptors are aligned. **f** Close-up view of the ligand-binding pocket. Concentration-response curves for D1R WT and the S198G mutant activated by tavapadon measured using the NanoBiT (**g**) and TANGO (**h**) β-arrestin recruitment assay and the cAMP accumulation assay (**i**). Each data point represents mean ± SEM from three independent experiments. Source data are provided as a Source Date file.

SKF83959 is substituted with chloride, shows a complete loss of potency in terms of both G protein and arrestin coupling due to the closer distance between TM5 and the larger chloride atom, which restricts the inward movement of TM5 and further activation of the receptor. To minimize the steric effect between TM5 and tavapadon, we substituted S198 into glycine residue with no side chain. As expected, S198G mutation resulted in an increase in efficacy of tavapadon for β-arrestin coupling, as indicated by the significantly increased maximum responses ($E_{max}$) (Fig. 3g, h).

By contrast, S198G mutation reduced the $E_{max}$ of tavapadon for G protein coupling (Fig. 3i). As S198 makes a hydrogen bond with dopamine, S198G mutation significantly reduced the potency of dopamine for $G_s$ coupling and β-arrestin recruitment (Supplementary Fig. 5a–c). To better quantify the biased signaling of D1R, we calculated the bias factor of tavapadon by using dopamine as a reference ligand (Supplementary Fig. 5d). In contrast to D1R WT where tavapadon shows no signaling bias, tavapadon is at least 100-fold more potent in β-arrestin recruitment than Gs activation for

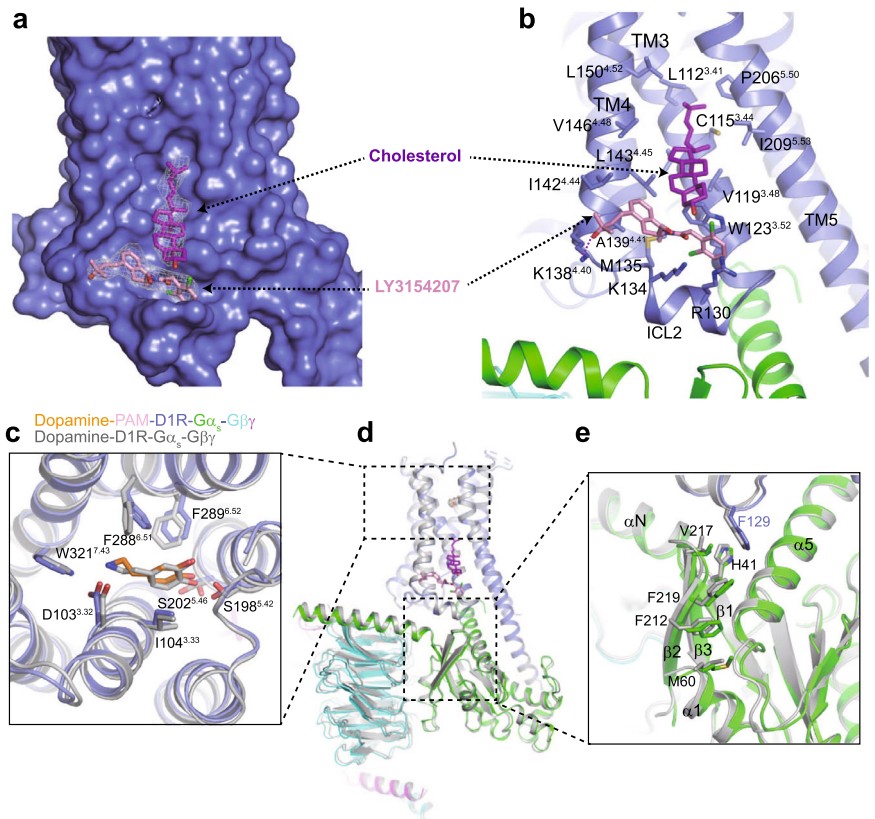

**Fig. 4 Allosteric modulation of D1R by PAM. a** Surface representation of D1R. EM densities for LY3154207 and cholesterol are shown. **b** Detailed interactions between LY3154207, cholesterol, and the receptor. Residues involved in binding are shown as sticks. **c–e** Structural comparison of the dopamine-LY3154207-bound D1R with D1R bound to dopamine alone.

the D1R S198G mutant. Taken together, these data suggest that the interaction between TM5 and the ligand is critical for determining the degree of overall signaling response and biased signaling of D1R.

**Allosteric modulation.** LY3154207 is a D1R-selective PAM that potentiates endogenous dopamine signaling. The structure of D1R bound to both dopamine and LY3154207 revealed two clear discontinuous densities perpendicular to each other, which are situated on the inner surface of the receptor in a pocket created by TM3, TM4, TM5, and ICL2 (Fig. 4a and Supplementary Fig. 6). The density parallel to the transmembrane helices could partially be resolved in the fenoldopam-bound structures (Supplementary Fig. 7a), suggesting that it may be derived from an endogenous lipid. Cholesterol can be modeled in this density due to its unique feature, a long aliphatic tail and a four-fused-ring structure, and LY3154207 is well fitted into the density that lies right above ICL2 and between TM3 and TM5 (Supplementary Fig. 7a). The LY3154207 binding site is consistent with that described in a recently published study[32] (Supplementary Fig. 7c). However, we can not exclude the possibility that the density for cholesterol is derived from a second LY3154207[33] owing to their similar shape and the modest map resolution (Supplementary Fig. 7d, e).

The dichlorophenyl ring of LY3154207 is sandwiched by R130[ICL2] and W123[3.52], and the remaining part of LY3154207 forms extensive hydrophobic interactions and Van der Waals contacts with hydrophobic residues including M135[ICL2], A139[4.41], I142[4.44], and L143[4.45], and the aliphatic parts of K134 and K138 on ICL2, both of which engage in hydrogen

bond interactions with LY3154207 (Fig. 4b). This PAM is further stabilized by cholesterol, which sits above it and lies in the hydrophobic cleft between TM3, TM4, and TM5. In support of the observations from structural analysis, recent studies have indicated that mutations of residues involved in either LY3154207 or cholesterol binding reduce or almost abolish the positive allosteric effects of LY3154207 on the potency of agonists[32,33]. Comparison of the structure of LY3154207-dopamine-bound D1R with that of D1R bound to dopamine alone[35] revealed that LY3154207 has little effect on the binding pose of dopamine and the conformation of residues in the ligand-binding pocket with minor side chain movement (Fig. 4c). Instead, it stabilizes the alpha-helical structure of ICL2 through interaction with R130 and K134 in ICL2 (Fig. 4b, d), which forms a random coil structure in the inactive receptor. The alpha-helical structure of ICL2 orients F129[ICL2] for efficient interaction with a hydrophobic pocket on Gα_s (Fig. 4e) and enhances the chance of D1R to adopt active conformations[50], accounting for its increased binding affinity for agonists upon LY3154207 binding. Moreover, αN of Gα moves downward opposite to the membrane, and β2 and β3 of Gα move toward α5, creating a narrower binding pocket where F129[ICL2] of the receptor is inserted (Fig. 4e). The αN-β1 hinge region serves as an allosteric link between ICL2 and the P loop of the nucleotide binding sites[47]. The further conformation change of this region caused by LY3154207 binding likely promotes the exchange of GDP for GTP. Taken together, LY3154207 and cholesterol interact with each other, and thus cooperatively potentiate the actions of agonists at receptors by stabilizing the active state of receptors, and possibly promoting G protein recruitment and the nucleotide exchange.

## Discussion

In this paper, we report the cryo-EM structure of the fenoldopam-bound D1R-G protein complex, revealing a ligand-binding mode of GPCRs where two identical agonists interact with each other and bind to distinct binding pockets, one in the OBP and the other in the EBP. In contrast, recently published studies reported the existence of only one fenoldopam in the OBP of D1R[33]. The distinct ligand-binding modes reported by our study and previous studies are attributed to the conformational differences of the receptor-G protein interface. Because of allosteric coupling between the ligand-binding pocket and the G protein-coupling interface, a subtle conformational change of the cytoplasmic pocket of the receptor can induce structural changes in the ligand-binding pocket, which subsequently influence ligand-binding, and vice versa. The dopamine binding mode and the conformation of the receptor-G protein interface also differ among studies conducted by different groups[32,33,35]. The different agonist binding modes indicate that agonists are highly dynamic in the receptors, which could lead to structurally and functionally distinct active states.

The structure of the tavapadon-bound D1R-G protein complex provides mechanistic insight into biased agonism. The distance between the ligand and S198$^{5.42}$ in TM5 of D1R is important for both ligand efficiency and biased signaling. Recently, it was shown that the distance between formoterol and S215$^{5.46}$ in TM5 in the structure of the arrestin-bound β1-adrenoceptor (β1AR) is increased, resulting in a narrower cytoplasmic pocket of the receptor due to the further inward movement of the cytoplasmic ends of TM5 and TM6 compared with that in a nanobody (G$_s$ protein mimetic)-bound β1AR[51]. Therefore, the closer distance between PW0464 and S198$^{5.42}$ in TM5 due to the extra oxygen atom in tavapadon is likely to interfere with the conformational state of D1R required for efficient coupling to β-arrestin. Indeed, S198G mutation that minimizes the steric effect between TM5 and the ligand leads to more β-arrestin biased. Clearly, structural characterization of the D1R-arrestin complex is required to fully understand the mechanism underlying biased signaling, and thus guide the design of drugs that could bias D1R signaling to achieve desirable therapeutic outcomes.

The structure of the D1R-G$_s$ complex simultaneously bound to dopamine and the PAM LY3154207 reveals that LY3154207 binds to the receptor's inner surface above ICL2, which is similar to location of the allosteric sites identified in β2AR for binding Cmpd-6FA[50]. In addition, a cholesterol molecule was found to interact with LY3154207 and further enhance its binding to the receptor. This cholesterol binding site in D1R is close to the site identified in the inactive kappa opioid receptor[52]. Moreover, we identify another cholesterol in the cleft between TM2, TM3, and TM4 of D1R, which is found in all structures of the D1R-G$_s$ complex (Supplementary Fig. 7f) and is identified at similar site in many GPCRs including β2AR[53]. The functional role of these cholesterol molecules warrants further investigation. In addition to the allosteric sites found on the surface of D1R, the second fenoldopam in the EBP adjacent to the OBP can be considered as an allosteric site. In fact, the extracellular vestibule adjacent to the OBP in the M2 muscarinic acetylcholine receptors has been targeted by a PAM named LY2119620[54]. Together, our structures of the D1R-G$_s$ complex bound to various ligands provide multiple templates for the rational design of D1R-selective agonists and PAMs.

## Methods

**Cloning and protein expression**. The human WT full-length D1R and mini-G$_s$399 were expressed as a fusion protein with a 3C protease site between them. The hemagglutinin (HA) signal peptide followed by a Flag tag was added into the N-terminus of D1R. Plasmids expressing the D1R-mini-G$_s$ fusion protein were transfected into Expi293F cells (Thermo Fisher Scientific) in SMM 293-TII expression medium (SinoBiological) using polyethyleneimine max (PEI, Polysciences), when cell density reached 1.5 million per ml. After shaking for 18 h, culture cells were supplemented with 5 mM sodium butyrate and 3 mM valproic acid, and were shaken for another 30 h before harvest.

For the fenoldopam-bound D1R-mini-G$_{\alpha s}$ fusion protein purification, cells were harvested by centrifugation at $1000 \times g$ for 10 min and lysed in hypotonic buffer (25 mM HEPES-NaOH pH 7.6, 50 mM NaCl, and 10 μM fenoldopam) using a glass dounce tissue grinder. The membrane fraction was collected by centrifugation and solubilized in buffer containing 25 mM HEPES pH 7.6, 150 mM NaCl, 0.5% LMNG (Anatrace), 0.1% cholesteryl hemisuccinate (CHS, Anatrace), 2 mM CaCl$_2$, and 10 μM fenoldopam (Targetmol) for 2 h at 4 °C. The solubilized protein solution was clarified by centrifugation and loaded onto anti-FLAG antibody affinity resin. After extensive washing with wash buffer containing 25 mM HEPES pH 7.6, 150 mM NaCl, 0.01% LMNG, 0.002% CHS, 2 mM CaCl$_2$, 10 mM MgCl$_2$, 2 mM KCl, 2 mM adenosine triphosphate, and 10 μM fenoldopam, protein was eluted in elution buffer (25 mM HEPES pH 7.6, 150 mM NaCl, 0.01% LMNG, 0.002% CHS, 5 mM EDTA, 0.1 mg/ml Flag peptide, 10 μM fenoldopam). The purified protein was incubated with PNGaseF (New England Biolabs) overnight to remove glycosylation.

To assemble the D1R-mini-G$_s$-Nb35 complex, purified D1R-mini-G$_{\alpha s}$ fusion protein was mixed in a 1:1.2:1.2 molar ratio with Gβ$_1$γ$_2$ harboring a C68S mutation and Nb35, which were purified as previously described[36,55]. The excess Gβ$_1$γ$_2$ and Nb35 proteins were removed using a Superose 6 10/300 column in buffer containing 25 mM HEPES pH 7.6, 150 mM NaCl, 0.01% LMNG, 0.002% CHS, and 10 μM fenoldopam. The fractions containing the complex were combined and concentrated to about 4 mg/ml for EM analysis.

For preparation of the tavapadon-bound and LY3154207 and dopamine-bound D1R-mini-G$_s$-Nb35 complexes, the procedures were performed as above except that fenoldopam was replaced by tavapadon (MedChemExpress), and LY3154207 (MedChemExpress) and dopamine (Sigma-Aldrich), respectively.

**Cryo-EM sample preparation and data collection**. To prepare cryo-EM grids, 3.0 μl of sample was added to a glow-charged 300 mesh holey carbon grid (Quantifoil Au R1.2/1.3). Excess sample was removed by blotting the grids for 4.0 s at a blotting force of 4 before plunge-freezing in liquid ethane using a Vitrobot MarkIV (Thermo Fisher Scientific) maintained at 8 °C and 100% humidity. Cryo-EM images were collected on a Titan Krios microscope (Thermo Fisher Scientific) at 300 kV using a BioQuantum GIF/K3 direct electron detector (Gatan) in superresolution mode. Images were recorded at a nominal magnification of ×64,000 with the defocus value set at 1.8 μm. Each movie stack was dose fractionated to 32 frames with a total dose of 50 e−/Å$^2$ for 2.56 s. Data collection parameters are summarized in Supplementary Table 1.

**Data processing**. For the fenoldopam-D1R-mini-G$_s$-Nb35 complex, a total of 1035 movie stacks were collected, gain normalized, motion corrected, dose weighted, and 2 × binned to a pixel size of 1.087 Å using MotionCor2[56]. Contrast transfer function (CTF) parameters were estimated using patch-based CTF estimation in cryoSPARC[57]. A total of 1,791,079 particles were auto-picked using Blob picker and subjected to two rounds of 2D classification in cryoSPARC to generate 454,215 good particles. Ab initio reconstruction and heterogeneous refinement in four classes were performed in cryoSPARC. One class that showed better density in the transmembrane domain was subjected to non-uniform refinement in cryoS-PARC, resulting in a map with a global resolution of 3.2 Å.

For the tavapadon-bound protein complex, similar data processing procedures were performed as above. A total of 1861 movies were collected, and 1,964,454 particles were picked using template picker in cryoSPARC with templates generated from the fenoldopam-bound complex. After two rounds of 2D classification, 581,994 particles with clear secondary structure features were selected and subjected to ab initio reconstruction in three classes in cryoSPARC. One class with clear features was applied to non-uniform refinement in cryoSPARC to yield a map at 3.3 Å.

For the dopamine-LY3154207-bound D1R complex, 824 movies were collected and 1,493,642 particles were picked using template picker in cryoSPARC. A total of 564,012 particles selected from two rounds of 2D classification were subjected to ab initio reconstruction and heterogeneous refinement with five classes. Two classes that had better density in the transmembrane domain were selected and run through non-uniform refinement to generate a map at 3.0 Å.

**Model building**. The structure of the dopamine-bound D1R-mini-G$_s$-Nb35 complex[35] was docked into EM density maps in Chimera[58]. Initial coordinates and refinement parameters for fenoldopam, tavapadon, LY3154207, cholesterol, and dopamine were prepared using eLBOW in PHENIX[59]. These small molecules were modeled into the respective EM density maps using COOT[60]. The model was manually built in COOT and refined in *Phenix.real_space_refinement* using reference structure and secondary structure restraints. Molprobity[61] and EMRinger[62] in PHENIX were used to evaluate the final models. The statistics for structure refinement are included in Supplementary Table 1. EM map and structure figures are prepared with ChimeraX and Pymol, respectively.

**cAMP accumulation assay**. The human WT D1R was cloned into the pcDNA3.1(+) vector, and D1R mutants were generated using the QuikChange method. HEK293 cells stably expressing the GloSensor biosensor (Promega) were cultured in a six-well plate in Dulbecco's modified Eagle's medium (DMEM, Gibco) supplemented with 10% fetal bovine serum (FBS, Gibco), penicillin and streptomycin and transfected with D1R plasmids using Lipo3000 (Thermo Fisher Scientific). Cells were incubated at 37 °C with 5% $CO_2$ for 24 h before being seeded in a white and clear-bottom 96-well plate. After incubation for another 24 h, the culture medium was exchanged to the equilibration medium containing $CO_2$-independent medium, 10% FBS and 1% D-luciferin. Cells were incubated at room temperature for 2 h before treatment with an increasing concentration of ligand. The luminescence signal was measured in 10 min after addition of ligands.

The cAMP accumulation assay was also performed in Expi293F cells transiently expressing the GloSensor biosensor. The plasmids expressing D1R or mutants and the GloSensor biosensor were co-transfected into Expi293F cells at a density of 1.5 million per ml using PEI max. After shaking for 24 h, cells were centrifuged and resuspended in HBSS reaction buffer (HBSS supplemented with 0.01% BSA and 5 mM HEPES, pH 7.4, 100 μg/ml D-luciferin). Cells were seeded into a 96-well plate in 95 μl of HBSS reaction buffer at a density of 1 million per ml. After incubation for 2 h at RT, ligands were series diluted in HBSS reaction buffer, and 5 μl of stock solution with titrated concentrations was added to each well. The luminescence signal was measured and plotted as a function of ligand concentration using non-linear regression with GraphPad Prism 8 (GraphPad Software). Three independent experiments, each in triplicate, were performed for each measurement. Significance was calculated by two-tailed Student's t test.

**NanoBiT mini-$G_s$ recruitment assay**. For monitoring the recruitment of mini-$G_s$ by ligand-activated D1R, we used NanoBiT mini-$G_s$ recruitment assay[37,42] in which interaction between D1R and mini-Gs was monitored by an NanoLuc-based enzyme complementation system named NanoBiT. A large fragment (LgBiT) and a small fragment (SmBiT) were fused to the N-terminus of mini-$G_s$ and the C-terminus of D1R to generate LgBiT-mini-$G_s$ and D1R-SmBiT fusion proteins, respectively. Expi293F were seeded into a six-well plate at a density of 1.5 million per ml and transfected with D1R-SmBiT or D1R mutants-SmBiT and LgBiT-mini-$G_s$ using PEI max. After 24 h, cells were resuspended in HBSS reaction buffer before being seeded into a 96-well plate in 95 μl of HBSS reaction buffer. After incubation for 2 h at room temperature, fenoldopam stock solutions were series diluted in HBSS reaction buffer, and 5 μl of stock solution with titrated concentrations was added to each well. The luminescence signal was measured in 5 min after addition of fenoldopam and normalized over baseline signal. The fold-changes were plotted as a function of fenoldopam concentration using non-linear regression with Prism. All experimental data were repeated for three independent times, each in triplicate. Significance was calculated by a two-tailed Student's t test.

**Microscale thermophoresis**. The D1R-mini-$G\alpha_s$ WT and K81V mutant were purified and prepared as mentioned above and prepared in buffer containing 25 mM HEPES pH 7.6, 150 mM NaCl, 0.03% DDM (Anatrace). Proteins were labeled with fluorescence using a Protein Labeling Kit RED-NHS (NanoTempler Technologies) according to manufacturer's instruction. The labeled protein was diluted to a final concentration of 50 nM with the binding buffer. In total, 10 μl of 100 μM fenoldopam was diluted 1:1 in 10 μl binding buffer to make a 16-sample serial dilution starting from 50 μM to 1.52 nM. The labeled protein was mixed with the equal volume of fenoldopam with 16 different concentrations at RT. After samples were filled into capillaries, all measurements were performed at 20% excitation power and 40% MST power with laser off/on times of 0 and 20 s in the Monolith NT.115 instrument. All experimental data were repeated in three independent times and analyzed by MO Affinity analysis software. The dissociation constants were determined using the "Kd model" for data fitting.

**TANGO β-arrestin recruitment assay**. PRESTO-TANGO assay was performed as previously described[43]. HTLA cells (a HEK293 cell line stably expressing a tTA-dependent luciferase reporter and a β-arrestin2-TEV fusion gene) were plated into a six-well plate and incubated in DMEM medium supplemented with 10% FBS, 100 U/ml penicillin and 100 μg/ml streptomycin, 2 μg/ml puromycin and 100 μg/ml hygromycin B at 37 °C with 5% $CO_2$. After incubation for 24 h, the plasmids expressing D1R or mutants were transfected with PEI max. Following 2 days incubation, transfected cells were transferred into poly-D-lysine-coated 96-well white, clear-bottomed plates (Corning) with 50,000 cells per well in 100 μl culture medium. Cells were incubated for another 24 h before treatment with drugs. After treatment for around 24 h, medium was replaced by the reaction buffer (HBSS buffer and 500 μg/ml D-luciferin). Luminescence was counted in Tecan Spark machine. Relative luminescence units were exported and plotted as a function of ligand concentration using a non-linear regression fit in GraphPad Prism 8. All experimental data were repeated for three independent times, each in duplicate or triplicate.

**NanoBiT β-arrestin recruitment assay**. HEK293 cells were seeded into 6-well plate 1 day before transfection. In total, 1 μg D1R-LgBit and 1 μg smBiT-β-arrestin

(an N-terminal smBiT) were co-transfected into cells using PEI max. After 1 day post transfection, cells were washed with D-PBS and resuspended in 3 ml HBSS reaction buffer. Cells were seeded into 96-well-plate with 0.1 million cells per well in 95 μl solution. After incubation at RT for 1 h, background luminescent signals were measured using luminescent microplate reader (Tecan, spark). In total, 5 μl of titrated concentrations of ligands were added and luminescent signals were measured 3–5 min after ligand addition and normalized to vehicle treatment. The normalized signals were fitted to a three-parameter sigmoidal concentration-response in Prism 8 software.

**Quantification of ligand bias**. Ligand bias was quantified using the "equiactive comparison" approach[63]. A "bias factor" (β) is calculated as the logarithm of the ratio of the relative activities for tavapadon at the cAMP accumulation assay ($G_s$ pathway, P1) and NanoBiT β-arrestin recruitment assay (Arrestin pathway, P2) compared with dopamine as a reference:

$$\log\left(\left(\frac{E_{max,P1}\ EC_{50,P2}}{EC_{50,P1}\ E_{max,P2}}\right)_{Tavapadon} \times \left(\frac{E_{max,P2}\ EC_{50,P1}}{EC_{50,P2}\ E_{max,P1}}\right)_{Dopamine}\right).$$

$E_{max}$ is defined as maximum response, and $EC_{50}$ as the concentration of ligand producing 50% of Emax.

**Reporting summary**. Further information on research design is available in the Nature Research Reporting Summary linked to this article.

## Data availability

The atomic structures have been deposited at the Protein Data Bank (PDB) under the accession codes 7X2C, 7X2D, and 7X2F. The EM maps have been deposited at the Electron Microscopy Data Bank (EMDB) under the accession numbers EMD-32964, EMD-32965 and EMD-32966 Source data are provided with this paper.

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

## Acknowledgements

We thank Dr. Bryan L. Roth for providing the HTLA cell lines and the PRESTO-Tango GPCR Kit. We thank staff at Shuimu BioSciences for their help with cryo-EM data collection. All EM images were collected at Shuimu BioSciences. We thank the staff at the Tsinghua University Branch of the National Protein Science Facility (Beijing) for their technical support on the MST experiment. This work was supported by Chinese Ministry of Science and Technology, Beijing Municipal Science & Technology Commission (Z201100005320012) and Tsinghua University.

## Author contributions

X.T. purified and assembled the protein complex. X.T. and S.Z. collected cryo-EM data and performed cryo-EM data processing and model building. X.T. did the cAMP accumulation assay. Y.N. performed the NanoBiT mini-G recruitment assay. S.C. performed the arrestin recruitment assay. S.Z. wrote the manuscripts with helpful input from P.X., X.Y., and Z.S.

## Competing interests

The authors declare no competing interests.
