## [Peer Review File · Nature Communications]

Ligand recognition and biased agonism of the D1 dopamine receptorREVIEWER COMMENTS

Reviewer #1 (Remarks to the Author):

The study by Zheng and colleagues reports three cryo-EM structures of the D1 dopamine receptor (D1R) in complex with the stimulatory G protein (Gs) heterotrimer. The fenoldopam-bound D1R-miniGs-Nb35 structure reveals unusual binding of two fenoldopam molecules, one to the OBP and the other to the EBP. The tavapadon-bound D1R-miniGs-Nb35 structure suggests that the interaction between TM5 and the ligand is important for the biased signaling. The dopamine-LY3154207-bound D1R-miniGs-Nb35 structure shows that LY3154207 and a cholesterol molecule cooperatively stabilize the ICL2 in an alpha helical conformation to efficiently engage the G protein. Findings based on structures are interesting, but the cell-based functional assays are not well designed. More evidences are required.

Major comments:

-As mentioned by authors: "These conformational differences may arise from the different versions of G α s in the complexes," Figure 2d shows that mini-Gs contributes to a boarder space between ICL2 and fenoldopam in the OBP which can accommodate the second fenoldopam in the EBP compared with the published fenoldopam-bound D1R-Gs structure. It is worth mentioning that mini-Gs is a truncated Gs in contrast to the entire Gs heterotrimer, the results obtained by assembling the complex with mini-Gs may be artificial. Since many mini-Gs-bound D1R structures have been published, are there any difference between them and the fenoldopam-bound D1R-miniGs structure in the intracellular cleft and EBP? Besides, functional assays should be proved by mini-Gs, for example, what will happen of mutations in figure 2c in the mini-Gs recruitment assay? Does it has any difference between these two assays?

-Since dopamine is similar to fenoldopam, does dopamine have a second binding pose like fenoldopam in D1R? In the dopamine-LY3154207-bound D1R-miniGs-Nb35 structure, not many details focus on the dopamine binding pocket.

Additional comments:

-Figure 2e and extended data figure 3c show that K81V mutation has a larger effect on fenoldopam in β -arrestin recruitment compared with G protein coupling. What will happen of the same mutation on the endogenous ligand dopamine and tavapadon?

- The point mutation of K81 is not consistent in the paper, figure 2c is K81V and figure 3d is K81T, Why?

-Figure 3g and 3h contribute to the conclusion that the interaction between TM5 and the ligand is critical for determining the biased signaling of D1R. Claims of bias signaling cannot be made by simply comparing EC50 or Emax values. Proper bias factor calculations must be performed, using the endogenous ligand as reference, considering difference sensitivity and variability from assay to assay.

-Line 329: "In addition to the allosteric sites found on the surface of D1R, the second fenoldopam in the EBP adjacent to the OBP can be considered as an allosteric site." More experimental data is needed to confirm this conclusion.

-All of the functional data are based on cAMP accumulation assay and β -arrestin recruitment assay (amplification system). It would be better if a consistent conclusion can be drawn from non-amplification system, such as BRET system for Fig 2 and 3.

Reviewer #2 (Remarks to the Author):

The article by Xiao Teng et al. presents three high-resolution structures of the D1 dopamine receptor (D1R) bound to different ligands and reveals a biased agonism of D1R. It is well-written and the data in general support the conclusions. However, significant revisions are required before its acceptance for publication.

Major points:

1. The title "Structural basis for the biased agonism of the D1 dopamine receptor" seems not appropriate because only one structure is related to the biased agonism of the D1R.
2. It is surprising that there are two fenoldopam molecules binding to D1R and the authors are advised to perform both chemical and pharmacological assays to confirm their co-existence. Compared to the published structure, the extended binding pocket (EBP) is crucial for another fenoldopam molecule binding to the receptor. However, mutations in the EBP did not show significant cAMP changes according to Figure 2c. If there exist two molecules, one in the orthosteric binding pocket (OBP) and the other in the EBP, mutations of OBP would only cause a partial impairment on cAMP responses because the other molecule should still be active. Obviously, additional studies should be conducted to clarify this situation.

3. The structure of D1R bound to dopamine and a PAM (LY3154207) is similar to that published previously by two other groups. It is important to make appropriate comparisons and offer new insights beyond what is known.

4. Line 307: "S198 and S202 in D1R are important for both ligand efficiency and biased signaling", however, the authors only displayed the data of S198G mutation with no information of S202, which should be provided in the revision.

Minor points:

1. Line 265: "R130ECL2" should be revised to "R130ICL2".

2. Statistical analysis should be provided for all functional experiments such changes in cAMP levels.

3. The size-exclusion chromatography and SDS-PAGE/Coomassie blue staining analysis on the receptor complexes should be added to in the supplementary information.

Reviewer #3 (Remarks to the Author):

The paper is a most interesting read. It gives a very insightful description of several cryo-EM structures of the D1R and correlates them nicely to the efficacy data.

Unfortunately, I cannot judge the paper with regard to its scientific quality sufficiently for this journal since I'm a bit out of expertise.

The paper is very well written and illustrated and an exciting read.

I would suggest to describe clearly what is new and different from the other D1R structures described to date (done in the text, but not comprehensively).

Furthermore, a clearer correlation between structure-activity relationships (SARs) and the data obtained in this paper would be desirable and of interest. maybe even illustrated in a graphical way.

We thank the referees for their evaluation of our manuscripts and their suggestions. Below, reviewer comments are in black and our responses are in blue.

REVIEWER COMMENTS

Reviewer #1 (Remarks to the Author):

The study by Zheng and colleagues reports three cryo-EM structures of the D1 dopamine receptor (D1R) in complex with the stimulatory G protein (Gs) heterotrimer. The fenoldopam-bound D1R-miniGs-Nb35 structure reveals unusual binding of two fenoldopam molecules, one to the OBP and the other to the EBP. The tavapadon-bound D1R-miniGs-Nb35 structure suggests that the interaction between TM5 and the ligand is important for the biased signaling. The dopamine-LY3154207-bound D1R-miniGs-Nb35 structure shows that LY3154207 and a cholesterol molecule cooperatively stabilize the ICL2 in an alpha helical conformation to efficiently engage the G protein. Findings based on structures are interesting, but the cell-based functional assays are not well designed. More evidences are required.

Major comments:

-As mentioned by authors: "These conformational differences may arise from the different versions of G α s in the complexes," Figure 2d shows that mini-Gs contributes to a boarder space between ICL2 and fenoldopam in the OBP which can accommodate the second fenoldopam in the EBP compared with the published fenoldopam-bound D1R-Gs structure. It is worth mentioning that mini-Gs is a truncated Gs in contrast to the entire Gs heterotrimer, the results obtained by assembling the complex with mini-Gs may be artificial.

Response: We have discussed in the main text that the distinct conformations of D1R-G protein complex obtained by different groups could be attributed to different version of G protein (mini-Gs versus full-length Gs). However, we respectfully disagree that the conformation reported here is artificial. All these structures only provided a single, low-energy conformation. Previous studies have shown that GPCRs should be considered as highly dynamic systems that exist in multiple active conformations. We further showed that K81V mutation that possibly precluded the second fenoldopam binding in the EBP based on the MST binding assay (Extended Data Fig. 3c and 3d) reduced the potency of fenoldopam in Gs activation by about five-fold and almost completely abolished fenoldopam-induced β -arrestin recruitment (Fig. 2), suggesting that a "broad" conformation of the receptor stabilized by two fenoldopam is essential for β -arresting recruitment and also play a role in Gs activation.

Since many mini-Gs-bound D1R structures have been published, are there any difference between them and the fenoldopam-bound D1R-miniGs structure in the intracellular cleft and EBP?

Response: Comparison between fenoldopam- and apomorphine-bound (PDB: 7JVQ) D1R-mini-Gs revealed conformational differences of the agonist-binding pocket and intracellular cleft, which could be attributed to ligand differences.

Besides, functional assays should be proved by mini-Gs, for example, what will happen of mutations in figure 2c in the mini-Gs recruitment assay? Does it has any difference between these two assays?

Response: We have performed NanoBiT mini-Gs recruitment assay for most mutants (Fig. 2d). The results are largely consistent between the two assays.

-Since dopamine is similar to fenoldopam, does dopamine have a second binding pose like fenoldopam in D1R? In the dopamine-LY3154207-bound D1R-miniGs-Nb35 structure, not many details focus on the dopamine binding pocket.

Response: We did not observed a second binding pose of dopamine like fenoldopam in the structure of dopamine-LY3154207-bound D1R-miniGs-Nb35 structure as well as the dopamine-bound D1R-miniGs-Nb35 structure (companion paper). However, we can not exclude the possibility that a second dopamine binds in the EBP as does fenoldopam, which was not captured in our structural studies probably because of its high dynamic property and low affinity binding in the EBP. Further studies are warranted to address this.

Additional comments:

-Figure 2e and extended data figure 3c show that K81V mutation has a larger effect on fenoldopam in β -arrestin recruitment compared with G protein coupling. What will happen of the same mutation on the endogenous ligand dopamine and tavapadon?

Response: We agree it is interesting to know the effect of K81V mutation on endogenous dopamine and tavapadon. However, this would require extensive work to properly interpret the results and is beyond the scope of this study.

- The point mutation of K81 is not consistent in the paper, figure 2c is K81V and figure 3d is K81T, Why?

Response: We have performed the same assay using the K81V mutant and obtained similar result. The K81T data was changed to K81V for consistency.

-Figure 3g and 3h contribute to the conclusion that the interaction between TM5 and the ligand is critical for determining the biased signaling of D1R. Claims of bias signaling cannot be made by simply comparing EC50 or Emax values. Proper bias factor calculations must be performed, using the endogenous ligand as reference, considering difference sensitivity and variability from assay to assay.

Response: We have used the endogenous ligand as reference and calculated the bias factor. The results yield the same conclusion that the interaction between TM5 and the ligand is critical for determining the biased signaling.

-Line 329: "In addition to the allosteric sites found on the surface of D1R, the second fenoldopam in the EBP adjacent to the OBP can be considered as an allosteric site." More experimental data is needed to confirm this conclusion.

Response: A GPCR allosteric modulator is defined as a ligand that does not occupy the orthosteric binding site where the endogenous ligand dopamine binds in the case of D1R. We can draw the conclusion that the second fenoldopam in the EBP can be considered as an allosteric site from structural information.

-All of the functional data are based on cAMP accumulation assay and β -arrestin recruitment assay (amplification system). It would be better if a consistent conclusion can be drawn from non-amplification system, such as BRET system for Fig 2 and 3.

Response: We have performed both NanoBiT (similar to BRET system) and TANGO β -arrestin recruitment assay for Fig2 and 3 and obtained consistent results.

Reviewer #2 (Remarks to the Author):

The article by Xiao Teng et al. presents three high-resolution structures of the D1 dopamine receptor (D1R) bound to different ligands and reveals a biased agonism of D1R. It is well-written and the data in general support the conclusions. However, significant revisions are required before its acceptance for publication.

Major points:

1. The title "Structural basis for the biased agonism of the D1 dopamine receptor" seems not appropriate because only one structure is related to the biased agonism of the D1R.

Response: The fenoldopam structure is actually related to the biased agonism. We changed the title to "Ligand recognition and biased agonism of the D1 dopamine receptor".

2. It is surprising that there are two fenoldopam molecules binding to D1R and the authors are advised to perform both chemical and pharmacological assays to confirm their co-existence. Compared to the published structure, the extended binding pocket (EBP) is crucial for another fenoldopam molecule binding to the receptor. However, mutations in the EBP did not show significant cAMP changes according to Figure 2c.

Response: The lack of a radiolabeled fenoldopam molecule did not allow us to perform the classic radioligand binding assay. We have tried the isothermal titration calorimetry experiments, which did not work well for this case. Finally, we chose microscale thermophoresis (MST) binding experiment. Consistent with structures, the MST experiment showed a biphasic binding curve for the wild-type D1R, suggesting the coexistence of two fenoldopam. Mutation of K81 involved in binding fenoldopam in the EBP led to one binding site for fenoldopam. This mutation almost completely abolished fenoldopam-induced β -arrestin recruitment and reduced the potency of fenoldopam in Gs activation by about 5-fold, suggesting that the fenoldopam binding in the EBP is essential for β -arresting recruitment and also play a role in Gs activation.

If there exist two molecules, one in the orthosteric binding pocket (OBP) and the other in the EBP, mutations of OBP would only cause a partial impairment on cAMP responses because the other molecule should still be active. Obviously, additional studies should be conducted to clarify this situation.

Response: A small molecule that can bind to GPCR does not necessarily mean that it can activate the receptor. For example, allosteric modulators can not activate the GPCRs by themselves but instead influence the potency of orthosteric agonists. In this case, the fenoldopam in the EBP could be considered as an allosteric modulator. K81V mutation that precludes the fenoldopam binding in the EBP based on the MST binding assay did reduce the potency of fenoldopam in arrestin recruitment and Gs activation. D103V mutation that abolished the orthosteric agonist binding completely abolished cAMP responses.

3. The structure of D1R bound to dopamine and a PAM (LY3154207) is similar to that published previously by two other groups. It is important to make appropriate comparisons and offer new insights beyond what is known.

Response: Our structure of D1R bound to dopamine and a PAM (LY3154207) mainly addressed the inconsistency of the binding site of LY3154207 from previously published studies conducted by two other groups, and further showed that the endogenous cholesterol can stabilize LY3154207 binding.

4. Line 307: "S198 and S202 in D1R are important for both ligand efficiency and biased signaling", however, the authors only displayed the data of S198G mutation with no information of S202, which should be provided in the revision.

Response: I am sorry for the miswriting. Here, we want to emphasize that S198 that is extensively studied is important for both ligand efficiency and biased signaling. We have rewritten this sentence to "The distance between the ligand and S198^{5,42} in TM5 of D1R is important for both ligand efficiency and biased signaling".

Minor points:

1. Line 265: "R130ECL2" should be revised to "R130ICL2".

Corrected.

2. Statistical analysis should be provided for all functional experiments such as changes in cAMP levels.

Provided.

3. The size-exclusion chromatography and SDS-PAGE/Coomassie blue staining analysis on the receptor complexes should be added to in the supplementary information.

Response: The size-exclusion chromatography and SDS-PAG have been provided in our companion paper which is now posted in BioRxiv.

Reviewer #3 (Remarks to the Author):

The paper is a most interesting read. It gives a very insightful description of several cryo-EM structures of the D1R and correlates them nicely to the efficacy data.

Unfortunately, I cannot judge the paper with regard to its scientific quality sufficiently for this journal since I'm a bit out of expertise.

The paper is very well written and illustrated and an exciting read.

I would suggest to describe clearly what is new and different from the other D1R structures described to date (done in the text, but not comprehensively).

Response: We have compared our structure with several recently published crystal structure and cryo-EM structures of D1R-Gs complex (Extended Data Fig. 1; Fig. 2e-2f; Fig. 3e-3f).

Furthermore, a clearer correlation between structure-activity relationships (SARs) and the data obtained in this paper would be desirable and of interest. maybe even illustrated in a graphical way.

Response: We have added discussions about the correlation between SAR and our structures in the paragraph of "Receptor activation by non-catechol agonists".

REVIEWER COMMENTS

Reviewer #1 (Remarks to the Author):

In the rebuttal letter, the authors only addressed some of my comments, there are still some questions were not answered very clearly and satisfactorily.

-In the first point, although the biphasic binding curve of fenoldopam for the wild-type D1R can suggest the coexistence of the two fenoldopam, the MST assay is an indirect assay to illustrate the conclusion. Additionally, in the following question, since there is no evidence for dopamine's second binding pocket, how does dopamine perform in the MST binding experiment?

-In the fifth point, there is still no any functional data to answer my question.

-In the seventh point, supplementary figure 5b shows that S198G mutation has significantly impaired β -arrestin signaling of the endogenous ligand dopamine, thus, biased signaling of β -arrestin can not be claimed.

-In the eighth point, the statement of allosteric site is wrong. The second binding site of fenoldopam can not be considered as an allosteric site, it just an extended binding site from the orthosteric site, whereas, the allosteric site is the binding site that is distinct from the orthosteric site which can play a regulatory role.

Reviewer #2 (Remarks to the Author):

I thank the authors for addressing my comments. The authors have addressed the majority of the referees' comments in a satisfactory manner. The content of the revised manuscript is more intact and reads logically. However, some minor points should be noticed:

1. The written form of "LgBiT" and "SmBiT" is suggested to be consistent in the part of methods.
2. Line 514, the "Afte" ought to be corrected as "After".

Reviewer #1 (Remarks to the Author):

In the rebuttal letter, the authors only addressed some of my comments, there are still some questions were not answered very clearly and satisfactorily.

Thanks again for your comments and suggestions.

-In the first point, although the biphasic binding curve of fenoldopam for the wild-type D1R can suggest the coexistence of the two fenoldopam, the MST assay is an indirect assay to illustrate the conclusion. Additionally, in the following question, since there is no evidence for dopamine's second binding pocket, how does dopamine perform in the MST binding experiment?

Thanks for pointing this out. We agree that the MST assay is an indirect assay. However, the structural information together with MST assay indicates the existence of two fenoldopam binding in D1R.

In this paper, we are mainly focused on validating the coexistence of two fenoldopam binding observed in the structure and understanding the functional relevance of the second fenoldopam binding. Whether there is a second dopamine binding is beyond the scope of this study. However, we are happy to add the MST binding data for dopamine in the revised version (**Supplementary Fig. 3e and 3f**), but we will tone down the second dopamine binding since we do not have direct evidence for that.

MST assay for dopamine showed that K81V mutation reduced the binding affinity of dopamine by about 15-fold (See figure above). As K81 is distant from dopamine in OBP, it is not clear how K81V mutation affects the binding affinity of dopamine. One explanation could be that a second dopamine may bind in a similar way to fenoldopam in EBP, which is not captured in our structure likely due to its dynamic property. Further studies are required to clarify whether a second dopamine binding site exists in D1R.

-In the fifth point, there is still no any functional data to answer my question.

Consistent with the MST binding assay, K81V mutation reduced the potency of dopamine in Gs activation by about 10-fold and significantly reduced both the potency and efficacy of dopamine in β -arrestin recruitment (**Supplementary Fig. 3g and 3h**). Since K81 is involved in direct binding with tavapadon based on the structure, we expect that its mutation should have similar effect on Gs activation (Fig. 3d) and β -arrestin recruitment.

-In the seventh point, supplementary figure 5b shows that S198G mutation has significantly impaired β -arrestin signaling of the endogenous ligand dopamine, thus, biased signaling of β -arrestin can not be claimed.

Please see descriptions in the main text from line 259-271. Biased signaling of β -arrestin is claimed for tavapadon compared with dopamine for the D1R S198G mutant. S198G significantly impairs both the G protein and β -arrestin signaling of dopamine since S198 forms a hydrogen bond with dopamine. In contrast, S198G mutation increases the efficacy

(E_{max}) of tavapadon for β -arrestin coupling (Fig. 3g and 3h) but decreases its efficacy for Gs coupling (Fig. 3i). Therefore, it is reasonable to claim that tavapadon is biased toward β -arrestin signaling for D1R S198G mutant compared to dopamine (Bias factor calculation in supplemental Fig. 5 supports our conclusion).

-In the eighth point, the statement of allosteric site is wrong. The second binding site of fenoldopam can not be considered as an allosteric site, it just an extended binding site from the orthosteric site, whereas, the allosteric site is the binding site that is distinct from the orthosteric site which can play a regulatory role.

The second fenoldopam site indeed plays a regulator role in Gs and β -arrestin signaling, which is the main focus of our study. K81V mutation that precludes the second fenoldopam binding impairs both the Gs and β -arrestin signaling.

I thank the authors for addressing my comments. The authors have addressed the majority of the referees' comments in a satisfactory manner. The content of the revised manuscript is more intact and reads logically. However, some minor points should be noticed:

1. The written form of "LgBiT" and "SmBiT" is suggested to be consistent in the part of methods.

Corrected.

2. Line 514, the "Afte" ought to be corrected as "After".

Corrected.

REVIEWERS' COMMENTS

Reviewer #1 (Remarks to the Author):

I thank the authors for addressing my comments. The authors have addressed my comments.

REVIEWERS' COMMENTS

Reviewer #1 (Remarks to the Author):

I thank the authors for addressing my comments. The authors have addressed my comments.

Response: Thanks for your comments